# Cell-intrinsic ceramides determine T cell function during melanoma progression

Matthias Hose[1]*[†], Anne Günther[1][†], Eyad Naser[2], Fabian Schumacher[3], Tina Schönberger[4], Julia Falkenstein[1], Athanasios Papadamakis[1], Burkhard Kleuser[3], Katrin Anne Becker[2], Erich Gulbins[2], Adriana Haimovitz-Friedman[5], Jan Buer[1], Astrid M Westendorf[1], Wiebke Hansen[1]*

[1]Institute of Medical Microbiology, University Hospital Essen, University Duisburg-Essen, Essen, Germany; [2]Institute of Molecular Biology, University Hospital Essen, University Duisburg-Essen, Essen, Germany; [3]Institute of Pharmacy, Freie Universität Berlin, Berlin, Germany; [4]Institute of Physiology, University Hospital Essen, University Duisburg-Essen, Essen, Germany; [5]Memorial Sloan Kettering Cancer Center, New York City, United States

*For correspondence:
matthias.hose@uk-essen.de
(MH);
wiebke.hansen@uk-essen.de
(WH)

[†]These authors contributed
equally to this work

Competing interest: The authors
declare that no competing
interests exist.

Reviewing Editor: Michael L
Dustin, University of Oxford,
United Kingdom

## Abstract

Acid sphingomyelinase (Asm) and acid ceramidase (Ac) are parts of the sphingolipid metabolism. Asm hydrolyzes sphingomyelin to ceramide, which is further metabolized to sphingosine by Ac. Ceramide generates ceramide-enriched platforms that are involved in receptor clustering within cellular membranes. However, the impact of cell-intrinsic ceramide on T cell function is not well characterized. By using T cell-specific Asm- or Ac-deficient mice, with reduced or elevated ceramide levels in T cells, we identified ceramide to play a crucial role in T cell function in vitro and in vivo. T cell-specific ablation of Asm in $Smpd1^{fl/fl}/Cd4^{cre/+}$ (Asm/CD4cre) mice resulted in enhanced tumor progression associated with impaired T cell responses, whereas $Asah1^{fl/fl}/Cd4^{cre/+}$ (Ac/CD4cre) mice showed reduced tumor growth rates and elevated T cell activation compared to the respective controls upon tumor transplantation. Further in vitro analysis revealed that decreased ceramide content supports CD4$^+$ regulatory T cell differentiation and interferes with cytotoxic activity of CD8$^+$ T cells. In contrast, elevated ceramide concentration in CD8$^+$ T cells from Ac/CD4cre mice was associated with enhanced cytotoxic activity. Strikingly, ceramide co-localized with the T cell receptor (TCR) and CD3 in the membrane of stimulated T cells and phosphorylation of TCR signaling molecules was elevated in Ac-deficient T cells. Hence, our results indicate that modulation of ceramide levels, by interfering with the Asm or Ac activity has an effect on T cell differentiation and function and might therefore represent a novel therapeutic strategy for the treatment of T cell-dependent diseases such as tumorigenesis.

## Editor's evaluation

The authors have used mouse genetics to show that acid sphingomyelinase mediated generation of ceramide promotes T cell activation and tumour control, whereas acid ceramidase reduces ceramide levels and impairs T cell activation and tumour control. The results also show that ceramide is polarised toward the immunological synapse. The work will be of relevance to those studying the role of lipids in signaling reactions generally and specifically to fields of T cell activation and tumour immunology.

## Introduction

Sphingolipids are structural components and bioactive molecules of cellular membranes with different functions in cellular processes. One important member of the sphingolipid family is ceramide. Ceramide has the ability to form ceramide-enriched platforms within the plasma membrane (*Kolesnick et al., 2000*). These microdomains contribute to receptor clustering and other protein interactions and are thereby involved in several signaling pathways and important cellular processes including proliferation, migration, differentiation, and apoptosis (*Zhang et al., 2009*).

The enzyme acid sphingomyelinase (Asm) generates ceramide by hydrolyzing sphingomyelin, whereas acid ceramidase (Ac) converts ceramide into sphingosine (*Hannun and Obeid, 2018*). Dysregulations in enzyme activity of the ceramide metabolism can lead to severe diseases. Specifically, mutations in the *SMPD1* gene, encoding for Asm, resulting in loss of or reduced Asm activity, cause the lipid-storage disease Niemann-Pick type A and B (NPD) (*Schuchman and Wasserstein, 2015*). NPD type A patients suffer from severe neuronal symptoms, due to sphingomyelin accumulations in the central nervous system (*Ohno, 1995*). In contrast, elevated Asm activity is associated with the development of major depressive disorders (*Kornhuber et al., 2005*). Pharmacological inhibitors of Asm (functional inhibitors of Asm, FIASMA) are therefore used as antidepressant drugs (*Beckmann et al., 2014*). Loss of Ac activity leads to the development of Farber disease (FD). FD patients suffer from arthralgia, hepatosplenomegaly, and a general developmental delay (*Yu et al., 2018*).

In addition, alterations in the sphingolipid metabolism play an important role in other pathological disorders including different tumor entities (*Ogretmen and Hannun, 2004*). For instance, Asm-generated ceramide-enriched platforms are crucial for the CD95-mediated induction of apoptosis (*Grassmé et al., 2003*). Cancer cells may upregulate Ac expression, leading to reduced ceramide abundance and increased levels of pro-survival lipid sphingosine-1-phosphate (S-1-P) and thereby foster their survival (*Morad and Cabot, 2013*; *Baran et al., 2007*; *Flowers et al., 2012*). Therefore, targeting of Asm and Ac in experimental cancer therapies has shown anti-tumoral efficacy. For example, ionizing radiation induces the activity of Asm, leading to ceramide generation and apoptosis of cancer cells, but fails to do so in lymphoblasts from NPD patients, which lack Asm activity (*Santana et al., 1996*). In addition, *Mauhin et al., 2021* demonstrated most recently, in a retrospective study, that there was an elevated incidence for cancer in NPD patients. We previously observed an enhanced tumor growth rate of transplanted tumor cells in Asm-deficient mice as compared to wildtype (WT) mice. Investigation of the tumor microvasculature identified apoptosis-resistant endothelial cells in Asm-deficient mice as a driver of elevated tumor growth (*Garcia-Barros, 2003*). In various cancer cell lines, Ac facilitates proliferation by the degradation of ceramide (*Govindarajah et al., 2019*). Pharmacological inhibition of Ac has been described to be effective for the treatment of patients suffering from colorectal cancer (*Sakamoto et al., 2005*).

Emphasizing the important role of ceramide in the modulation of tumorigenesis, recent studies by Ghosh et al. provided evidence that the application of exogenous C2 ceramide induces a strong anti-tumor response by increasing frequencies of cytotoxic CD8+ and IFN-γ-producing CD4+ T cells (*Ghosh et al., 2020*). Among other immune cells, T cells play a crucial role during tumor progression. In melanoma patients, the ratio of cytotoxic CD8+ T cells versus CD4+Foxp3+ regulatory T cells (Tregs) in the tumor microenvironment is predictive for the disease outcome (*Jacobs et al., 2012*). The infiltration of Tregs into the tumor tissue is considered as a critical step during tumorigenesis. We provided evidence that Neuropilin-1, highly expressed by Tregs (*Bruder et al., 2004*), regulates the migration of Tregs into VEGF-producing tumor tissue accompanied by elevated tumor progression (*Hansen et al., 2012*). Depletion of Tregs improved the anti-tumoral immune response of CD8+ T cells in colitis-associated colon cancer, emphasizing the important role of the T cell composition in tumorigenesis (*Pastille et al., 2014*).

Analysis of immune responses in Asm-deficient mice has shown an impaired cytotoxic activity of CD8+ T cells during LCMV infection (*Herz et al., 2009*). Moreover, Asm has been identified as a negative regulator of Treg development (*Zhou et al., 2016*). Asm-deficient or FIASMA-treated mice showed increased numbers of Tregs in comparison to WT animals (*Hollmann et al., 2016*). In accordance, we detected lower Treg frequencies in spleens of t-Asm/CD4cre mice, which overexpress Asm specifically in T cells (*Hose et al., 2019*). In CD4+ T cells, isolated from human peripheral blood, inhibition of Asm led to an impaired T cell receptor (TCR) signal transduction accompanied by reduced T cell proliferation and impaired CD4+ T helper (Th) cell differentiation (*Bai et al., 2015*). The role of

Ac in T cell responses is largely unclear. Nevertheless, as mentioned above, several lines of evidence indicate that ceramide metabolism participates in the regulation of T cell responses.

Here, we demonstrate that elevated ceramide concentrations facilitate TCR signaling cascades and determine T cell activation and differentiation in vitro. Strikingly, transplantation of B16-F1 melanoma cells into *Smpd1*fl/fl/*Cd4*cre/+ (Asm/CD4cre) or *Asah1*fl/fl/*Cd4*cre/+ (Ac/CD4cre) mice, which exhibit decreased or increased ceramide levels in T cells, respectively, revealed that increased ceramide concentrations improve the anti-tumoral T cell response during melanoma progression.

## Results

### Asm-deficient mice show enhanced tumor growth accompanied by reduced T cell activation

To analyze the impact of Asm activity on T cell function during tumorigenesis, we transplanted B16-F1 melanoma cells into *Smpd1*-deficient mice (Asm-KO) or control (Asm-WT) littermates. We observed significantly accelerated tumor growth rates in Asm-deficient mice compared to WT mice as previously described (*Garcia-Barros, 2003*; *Figure 1A*). Subsequently, we analyzed T cell frequencies in tumor draining lymph nodes (dLN) and tumors as well as the activation status of tumor-infiltrating lymphocytes (TILs). We detected lower frequencies of CD4+ and CD8+ T cells in dLNs but increased frequencies of Tregs in tumor bearing Asm-deficient mice compared to WT mice (*Figure 1B*). Furthermore, CD4+ and CD8+ TILs showed a reduced expression of IFN-γ and CD44 (*Figure 1C*), indicating a decreased T cell response in Asm-deficient mice during tumorigenesis. These results suggest that elevated tumor growth in Asm-deficient mice correlates with an insufficient T cell response.

Moreover, we investigated whether pharmacological inhibition of Asm also affects the T cell response during tumorigenesis in a similar way. Therefore, we treated tumor-bearing C57BL/6 mice with amitriptyline by daily intraperitoneal(i.p.) injection. Indeed, we detected enhanced tumor progression in amitriptyline-treated mice compared to tumor-bearing mice that received the vehicle. This was accompanied by reduced activation of TILs in terms of IFN-γ and CD44 expression in amitriptyline-treated C57BL/6 mice compared to PBS-treated mice (*Figure 1—figure supplement 1A, B*).

Next, we analyzed the effect of Asm deficiency on the Foxp3+ Treg subpopulation in vitro, since we observed elevated frequencies of Tregs in dLN and tumors of tumor-bearing Asm-deficient mice. In accordance with results from *Hollmann et al., 2016* and our own study of Asm-overexpressing T cells (*Hose et al., 2019*), we detected elevated Treg frequencies in spleens of naïve Asm-KO mice as compared to controls (*Figure 1D*). Moreover, induction of Tregs in vitro revealed an improved capacity of CD4+CD25− T cells to differentiate into Tregs when Asm activity is absent (*Figure 1E*). In summary, these data provide evidence that ceramide generation by Asm activity in CD4+ T cells interferes with Treg differentiation.

### CD4+ T cell depletion in tumor-bearing Asm-deficient mice reveals CD8+ T cell dysfunction

Next, we asked whether the observed increase in relative numbers of Tregs contributes to the insufficient CD8+ T cell response accompanied by enhanced tumor growth in Asm-deficient mice. For this purpose, we depleted CD4+ T cells from Asm-deficient and Asm-WT mice and transplanted B16-F1 melanoma cells. Interestingly, depletion of CD4+ T cells reduced tumor growth in both Asm-WT and Asm-KO mice. Still, tumors of CD4+ T cell-depleted Asm-deficient mice showed higher tumor growth rates than CD4+ T cell-depleted Asm-WT mice (*Figure 2A*). Remarkably, depletion of CD4+ T cells in Asm-WT mice abolished tumor progression almost completely. Using flow cytometry analysis, we confirmed the successful depletion of CD4+ T cells in dLN and tumor tissue and observed an increase of CD8+ T cell frequencies and numbers, which was significantly less pronounced in dLNs and tumors of Asm-KO mice (*Figure 2B*). Strikingly, analysis of CD8+ TILs from Asm-deficient mice revealed reduced expression of activation-associated molecules like IFN-γ, CD44, and granzyme B as compared to Asm-WT mice after CD4+ T cell depletion (*Figure 2C*). These results indicate that Asm-deficiency interferes with CD8+ T cell responses in tumor-bearing mice independent of CD4+ T cells.

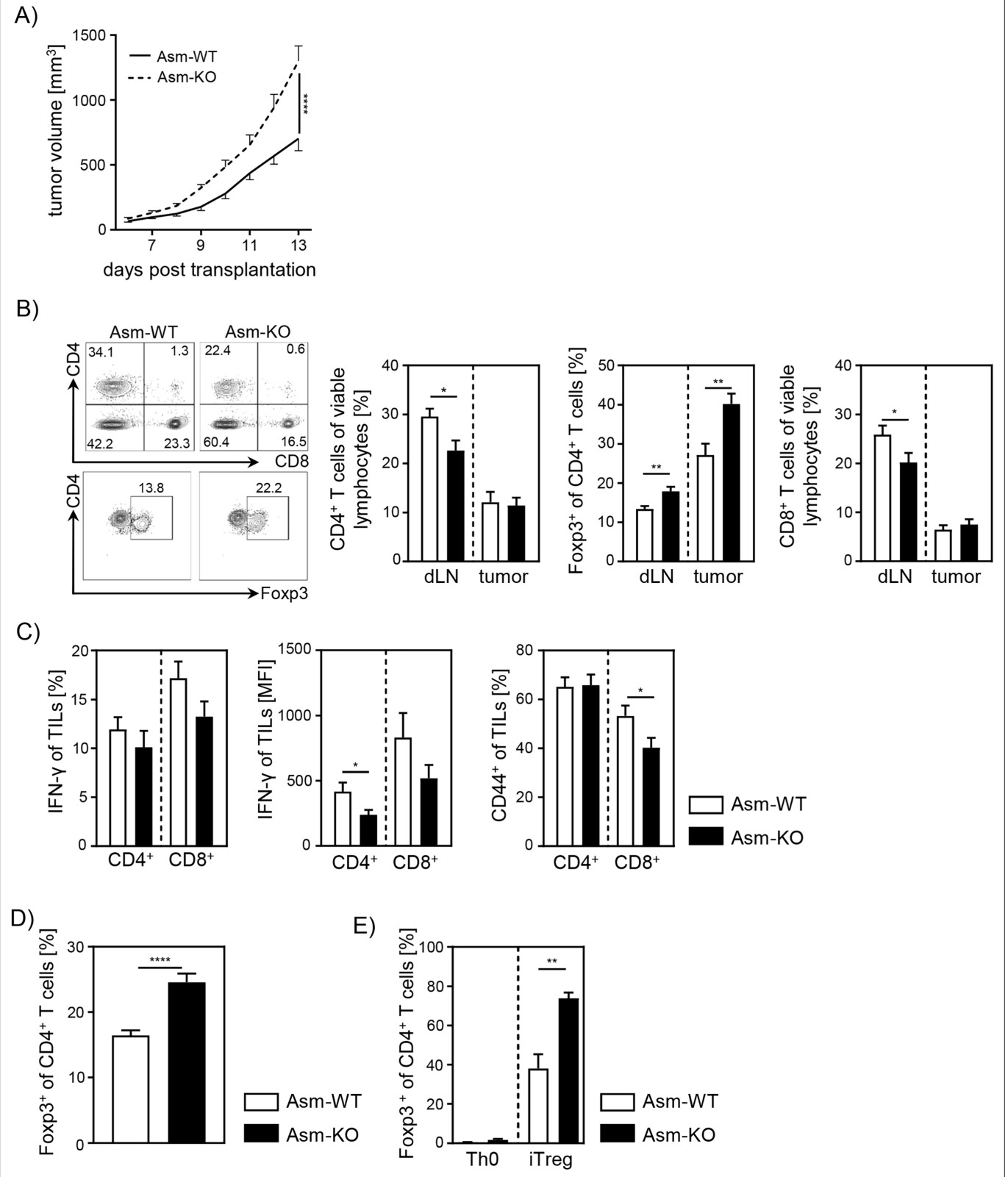

**Figure 1.** Ablation of acid sphingomyelinase (Asm) results in decreased T cell activation and enhanced tumor growth. (**A**) B16-F1 melanoma cells were transplanted into *Smpd1*-deficient mice (Asm-KO) mice and control littermates (Asm-WT). Tumor volume was monitored daily once tumors were palpable (n=12–18). (**B**) Frequencies of CD4+ T cells, CD4+Foxp3+ Tregs, and CD8+ T cells within draining lymph nodes (dLN) and tumor of Asm-KO and Asm-WT mice were determined by flow cytometry. Representative contour plots (dLN) are shown in the left panel. (**C**) IFN-γ, and CD44 expression of

*Figure 1 continued on next page*

*Figure 1 continued*

CD4[+] and CD8[+] tumor-infiltrating lymphocytes (TILs) in tumor-bearing mice. (**D**) Percentages of Foxp3[+] Tregs of CD4[+] T cells within spleen of naive Asm-KO and Asm-WT mice were determined by flow cytometry (n=16–17). (**E**) Sorted CD4[+]CD25[−] T cells from Asm-KO and Asm-WT mice were stimulated with anti-CD3 and anti-CD28 in the presence of IL-2 and TGF-β1 (iTreg). Respective controls (Th0) were only stimulated with anti-CD3 and anti-CD28 antibodies. After 3 days, Treg differentiation was analyzed by Foxp3 expression (n=3–4). Results from 2 to 4 independent experiments are depicted as mean ± SEM. Statistical analysis was performed by two-way ANOVA with Sidak's multiple comparisons or Student's t-test. (*p<0.05, **p<0.01, ****p<0.0001).

The online version of this article includes the following source data and figure supplement(s) for figure 1:

**Source data 1.** Ablation of Asm results in decreased T cell activation and enhanced tumor growth.

**Figure supplement 1.** Tumor growth and T cell response in amitriptyline-treated mice.

## T cell-specific ablation of Asm reduces CD8[+] T cell activation in vitro

To gain further insights into the role of cell-intrinsic Asm activity in CD8[+] T cells, we analyzed the phenotype and function of Asm-deficient CD8[+] T cells in vitro. To exclude an impact of other Asm-deficient cells present in Asm-KO mice, we made use of *Smpd1*[fl/fl]/*Cd4*[cre/+] (Asm/CD4cre) mice, which lack Asm expression specifically in T cells (*Figure 3A*, *Figure 3—figure supplement 1A*). This results in significantly reduced ceramide levels in unstimulated, as well as anti-CD3/anti-CD28 stimulated CD8[+] and CD4[+] T cells compared to T cells from WT littermates (*Smpd1*[fl/fl]/*Cd4*[+/+]) (*Figure 3B*, *Figure 3—figure supplement 1B*). Consistent with decreased T cell activation observed in tumor-bearing Asm-KO mice, in vitro stimulation of sorted CD8[+] T cells from Asm/CD4cre mice led to reduced expression of early T cell activation-associated molecules. This was reflected by lower CD25, CD69, and CD44 expression among Asm-deficient CD8[+] T cells compared to Asm-proficient CD8[+] T cells after 24 hr of stimulation (*Figure 3C*). Moreover, co-cultivation of antigen-specific cytotoxic lymphocytes (CTLs), generated from Asm/CD4cre/OT-I mice, together with ovalbumin(OVA)-loaded target cells revealed a reduced killing capacity of Asm-deficient CD8[+] T cells (*Figure 3D*). Well in line, CD8[+] T cells from Asm/CD4cre mice showed reduced granzyme B expression in response to TCR stimulation (*Figure 3E*). Strikingly, this phenotype was partially rescued by the addition of exogenous C16 ceramide during stimulation (*Figure 3F*). Altogether, these data indicate that ceramide is important for effective CD8[+] T cell responses, at least in vitro. In addition to the effect of ceramide on CD8[+] T cell function, we detected a reduced capacity of CD4[+] T cells from Asm/CD4cre mice to differentiate into Th1 cells, but to a higher extent into Tregs in vitro (*Figure 1—figure supplement 1C, D*), which is in line with our previous results from Asm-KO mice (*Figure 1E*).

## Cell-intrinsic Asm activity determines T cell activation during tumorigenesis

To investigate whether T cell-specific Asm ablation has also an impact on T cell responses during tumorigenesis in vivo, we transplanted B16-F1 melanoma cells into Asm/CD4cre and WT mice and observed significantly higher tumor growth rates in Asm/CD4cre mice compared to control mice (*Figure 4A*). Again, we analyzed T cell frequencies and T cell activation in dLNs and TILs. In line with results from Asm-KO mice (*Figure 1B*), we detected elevated percentages of tumor-infiltrating Foxp3[+] Tregs as well as reduced frequencies of CD4[+] and CD8[+] T cells in dLNs from Asm/CD4cre mice compared to control littermates (*Figure 4B*). Moreover, absolute cell numbers of intratumoral T cells were reduced in Asm/CD4cre mice compared to WT littermates. TILs from T cell-specific Asm-deficient mice showed decreased expression of IFN-γ and TNF-α, as well as granzyme B indicating a reduced anti-tumoral T cell response (*Figure 4C*). These results provide evidence that cell-intrinsic Asm activity has an impact on T cell responses in vitro and during ongoing immune responses in vivo.

## Ceramide accumulates at TCR synapse

The previous experiments indicate that ceramide generation by Asm activity is involved in T cell function. In order to elucidate the subcellular localization of ceramide, we performed fluorescence microscopy of CD8[+] T cells stimulated with CD3/CD28 MACSiBead particles and stained for ceramide, CD3, and TCR-beta. Indeed, ceramide accumulates at the contact site between T cell and particle. Strikingly, ceramide co-localizes with CD3 (*Figure 5A*) and TCR beta, respectively (*Figure 5B*). This co-localization of TCR and ceramide suggests an involvement ceramide in TCR signaling.

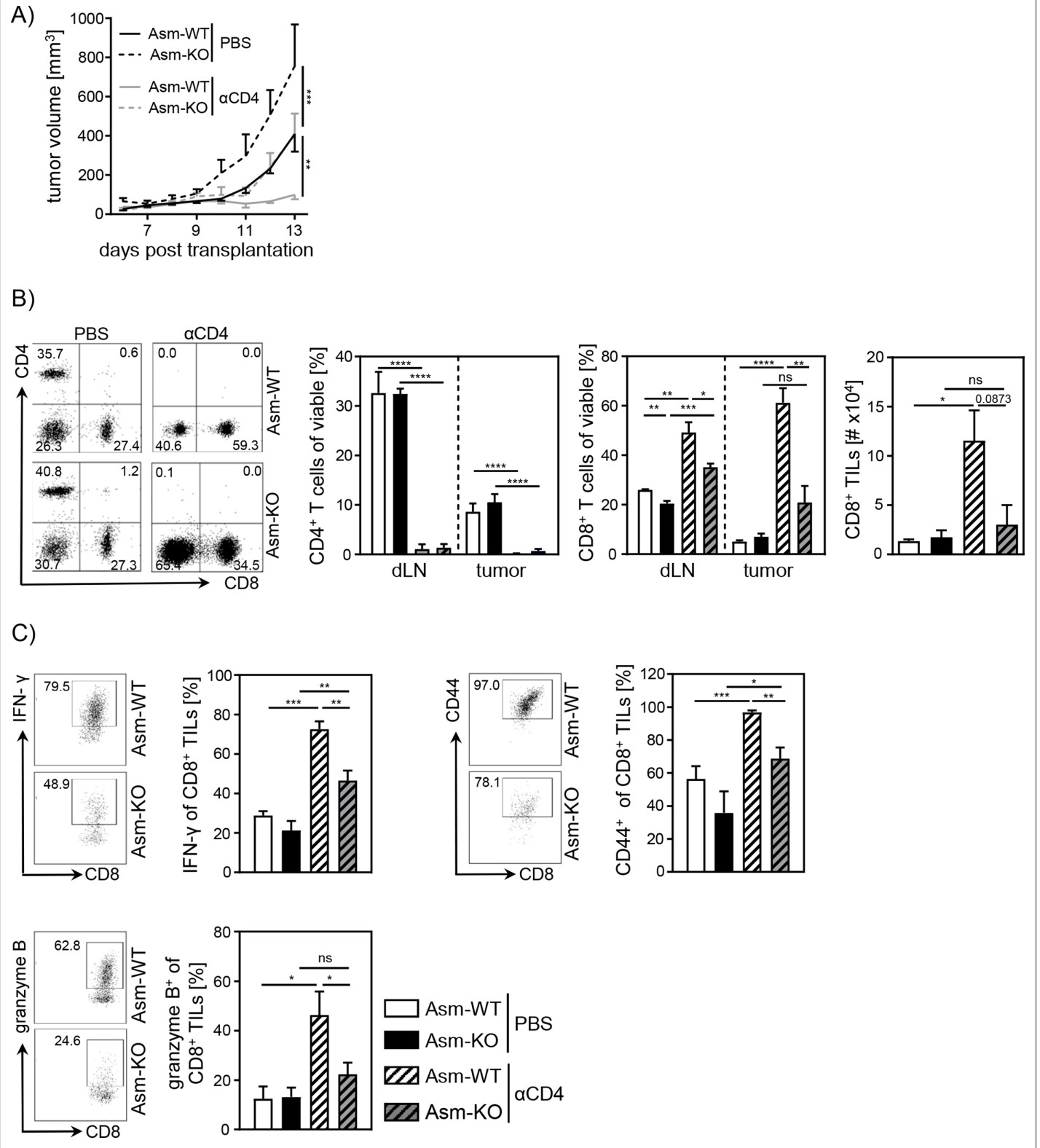

**Figure 2.** Impaired CD8[+] T cell function in acid sphingomyelinase (Asm)-deficient tumor-bearing mice upon CD4[+] T cell depletion. CD4[+] T cells were depleted from Asm-WT and Asm-KO mice by repeated i.p. injection of anti-CD4 depleting antibody. Control groups received PBS. B16-F1 tumor cells were transplanted 1 day later subcutaneously. (**A**) Tumor volume was determined after tumor establishment based on caliper measurements (n=3–7). (**B**) Frequencies of CD4[+] and CD8[+] T cells in dLN and tumor and (**C**) expression of IFN-γ, CD44, and granzyme B of CD8[+] tumor-infiltrating lymphocytes (TILs) were analyzed using flow cytometry. Representative dot plots are shown in the left panel. Results from two independent experiments are depicted

*Figure 2 continued on next page*

Figure 2 continued

as mean ± SEM. Statistical analysis was performed by two-way ANOVA with Sidak's multiple comparisons or Student's t-test. (*p<0.05, **p<0.01, ***p<0.001, ****p<0.0001).

The online version of this article includes the following source data for figure 2:

**Source data 1.** Impaired CD8+ T cell function in Asm-deficient tumor-bearing mice upon CD4+ T cell depletion.

## Elevated ceramide generation in Ac-deficient CD8+ T cells results in enhanced cytotoxic function in vitro

To further validate the important role of ceramide in CD8+ T cell responses, we made use of *Asah1*<sup>fl/fl</sup>/*Cd4*<sup>cre/+</sup> Ac/CD4cre mice. In these mice, T cells are deficient for Ac, the enzyme that catalyzes the hydrolysis of ceramide into sphingosine. As validated by qPCR, both CD8+ and CD4+ T cells lack Ac expression (*Figure 6A*, *Figure 6—figure supplement 1A*), resulting in elevated ceramide concentrations in unstimulated and anti-CD3/anti-CD28 T cells from Ac/CD4cre mice compared to control WT littermates (*Asah1*<sup>fl/fl</sup>/*Cd4*<sup>+/+</sup>) (*Figure 6B*, *Figure 6—figure supplement 1B*). Since we observed a co-localization between ceramide and CD3 and TCR-beta upon stimulation, respectively (*Figure 5*), we wondered whether elevated ceramide levels in Ac-deficient T cells might correlate with TCR signaling. Therefore, we isolated splenocytes from Ac/CD4cre and control mice and stimulated them with anti-CD3/anti-CD28 in vitro. Indeed, CD8+ and CD4+ T cells from Ac/CD4cre mice showed significantly elevated phosphorylation of the TCR signaling molecules ZAP70 and PLCγ compared to control CD8+ T cells (*Figure 6C*, *Figure 6—figure supplement 1C*). The elevated phosphorylation of ZAP70 was further confirmed by western blot analysis of isolated Ac-deficient CD8+ T cells (*Figure 6D*). Well in line with these data, stimulation of CD8+ T cells from Ac/CD4cre mice led to an increased expression of granzyme B compared to CD8+ T cells from control littermates (*Figure 6E*). Strikingly, CTLs from Ac/CD4cre/OT-I mice showed an improved killing capacity in comparison to control cells (*Figure 6F*). Although Ac ablation had no impact on Th1 differentiation in regard to IFN-γ expression, CD4+ T cells from Ac/CD4cre mice showed an enhanced expression of granzyme B under Th1-polarizing conditions in vitro (*Figure 6—figure supplement 1D*).

In this study, we could confirm that Asm or Ac deficiency alters the ceramide concentrations in T cells. In consequence, T cells from Asm/CD4cre mice showed reduced activation and killing capacity in vitro and in vivo, whereas T cells from Ac/CD4cre mice revealed elevated phosphorylation of TCR signaling molecules and an improved killing capacity in vitro. To investigate whether the abundance of T cell synaptic ceramide is affected in Asm-deficient or Ac-deficient T cells, we isolated CD8+ T cells from respective mouse strains and analyzed the subcellular localization of ceramide (*Figure 7A*). By calculating the ratio of the ceramide signal in the synapse to the signal at the back of the cells, we elucidated the polarization of ceramide toward the stimulating beads. Indeed, synaptic ceramide signal in Ac-deficient T cells was highly elevated compared to Asm-deficient T cells (*Figure 7B*), emphasizing that ceramide is involved in TCR clustering and the interaction of T cells with antigen-presenting cells (APCs).

## T cell-specific Ac ablation promotes anti-tumoral immune response

Since our data indicate an important role for cell-intrinsic ceramide in T cell activation in vitro, we next analyzed the effect of accumulating ceramide in T cells in vivo. Thus, we transplanted B16-F1 melanoma cells into Ac/CD4cre mice and control littermates. In contrast to Asm/CD4cre mice, Ac/CD4cre mice showed a significant reduction in tumor size compared to control mice (*Figure 8A*). Although mice did not differ regarding T cell frequencies or numbers (*Figure 8B*), we detected an elevated T cell activation in Ac/CD4cre tumor-bearing mice. This was reflected by increased IFN-γ and granzyme B expression of CD4+ and CD8+ TILs in comparison to control mice (*Figure 8C*). From these results, we conclude that enhancing cell-intrinsic ceramide by ablation of Ac activity promotes the T cell function, whereas reduced ceramide levels due to loss of Asm activity are accompanied by a decrease in the activity of T cells.

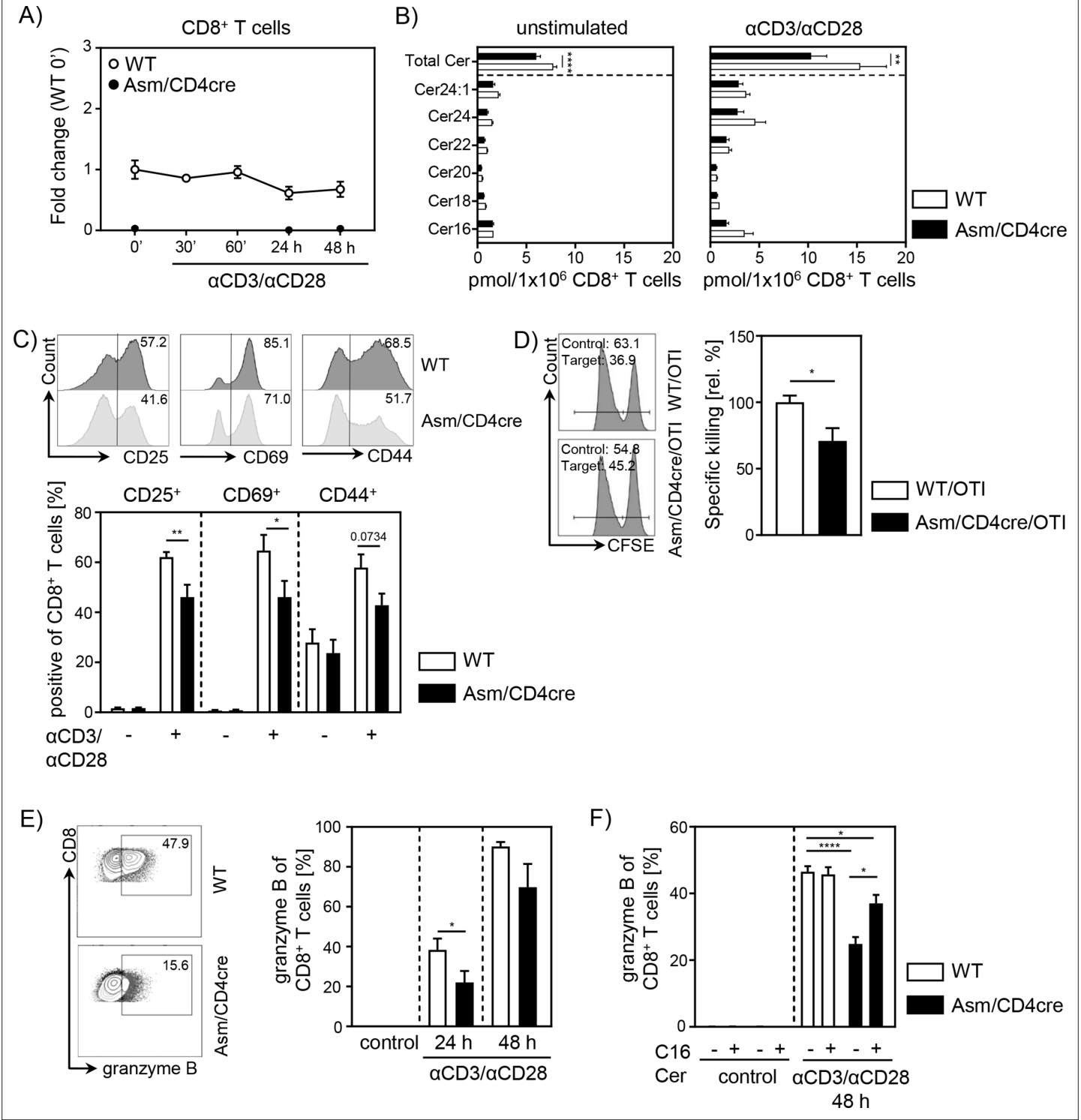

**Figure 3.** T cell-specific acid sphingomyelinase (Asm) deficiency leads to reduced CD8⁺ T cell activation in vitro. Isolated CD8⁺ T cells from *Smpd1^{fl/fl}/Cd4^{cre/+}* mice (Asm/CD4cre) and *Smpd1^{fl/fl}/Cd4^{+/+}* littermates (wildtype [WT]) where either left unstimulated or stimulated with anti-CD3 and anti-CD28 for indicated time points. (**A**) mRNA expression of Asm (*Smpd1*) following activation was analyzed by RT-qPCR (n=4–8). (**B**) Ceramide levels of CD8⁺ T cells were determined by mass spectrometry (n=4). (**C**) Expression of CD25, CD69, and CD44 was analyzed by flow cytometry (n=6–7). Representative histograms are shown in the upper panel. (**D**) OVA-specific cytotoxic lymphocytes were generated and incubated with OVA-peptide 257–264 loaded CFSE^high-labeled target and unloaded CFSE^low-labeled control cells. Specific killing was evaluated by frequencies of target and control populations determined by flow cytometry (n=6–7). Representative histograms are shown in the left panel. (**E**) Frequencies of granzyme B-expressing CD8⁺ T cells

*Figure 3 continued on next page*

*Figure 3 continued*

from Asm/CD4cre mice and WT littermates without and (**F**) in the presence of C16 ceramide were analyzed by flow cytometry (n=4–8). Representative contour plots are shown in the left panel. Results from two to four independent experiments are depicted as mean ± SEM. Statistical analysis was performed by two-way ANOVA with Sidak's multiple comparisons, Mann-Whitney U-test, or Student's t-test. (*p<0.05, **p<0.01, ****p<0.0001).

The online version of this article includes the following source data and figure supplement(s) for figure 3:

**Source data 1.** T cell-specific Asm deficiency leads to reduced CD8$^+$ T cell activation in vitro.

**Figure supplement 1.** In vitro characterization of acid sphingomyelinase (Asm)-deficient CD4$^+$ T cells.

## Discussion

In the present study, we investigated the impact of cell-intrinsic Asm and Ac activity on the phenotype and function of CD4$^+$ as well as CD8$^+$ T cells in vitro, and during tumorigenesis in vivo. We identified a correlation between ceramide levels and T cell activity. Reduced ceramide content due to loss of Asm activity triggered Treg induction and interfered with effector T cell responses, whereas elevated ceramide concentrations induced by ablation of Ac expression resulted in enhanced T cell activation. Strikingly, cell-intrinsic ceramide levels also correlated with anti-tumoral immune responses and tumor growth in T cell-specific Asm-deficient or Ac-deficient mice.

Several studies already described Asm as regulator of CD4$^+$ T cell function and differentiation (*Zhou et al., 2016*; *Hollmann et al., 2016*; *Bai et al., 2015*). Well in line, we demonstrated that cell-intrinsic Asm activity determines T cell activation and differentiation. However, the mechanism that triggers Asm activity in T cells is still controversial. *Mueller et al., 2014* postulated that CD28 stimulation induces Asm activity in T cells, whereas co-stimulation of CD3 and CD28 does not. In contrast, Asm activation has been described in isolated human CD4$^+$ T cells in response to CD3/CD28 co-stimulation (*Bai et al., 2015*), while Wiese et al. identified CD28 co-stimulation to be required for enhanced human CD4$^+$Foxp3$^+$ Treg frequencies upon Asm inhibition (*Wiese et al., 2021*). In accordance, we detected significantly elevated ceramide concentrations upon anti-CD3/anti-CD28 treatment of WT CD4$^+$ T cells, which were substantially reduced in Asm-deficient T cells, suggesting that co-stimulation of CD3/CD28 is important for the induction of Asm activity in CD4$^+$ T cells.

Others and we demonstrated that transplantation of melanoma cells into Asm-deficient mice results in accelerated tumor growth. As one underlying mechanism, apoptosis-resistant endothelial cells have been proposed (*Garcia-Barros et al., 2004*). Here, we provide evidence that Asm deficiency in T cells contributes to an impaired anti-tumoral immune response resulting in loss of tumor growth control. We detected elevated percentages of Foxp3$^+$ Tregs accompanied with reduced expression of activation-associated molecules of effector T cells from tumor-bearing Asm-deficient mice, amitriptyline-treated mice, and importantly, in T cell-specific Asm-deficient mice, in contrast to the respective controls. These results indicate that the T cell-intrinsic Asm activity regulates T cell function and differentiation most likely via the generation of ceramide.

It is well established that Tregs infiltrate the tumor tissue and interfere with an effective local immune response (*Hansen et al., 2012*; *Pastille et al., 2014*; *Jarnicki et al., 2006*; *Akeus et al., 2015*). Moreover, a higher ratio of Tregs to CD8$^+$ effector T cells correlates with poorer disease outcome in cancer (*Sato et al., 2005*; *Gao et al., 2007*; *Angelova et al., 2015*). To dissect whether enhanced Treg frequencies from Asm-deficient mice are responsible for the enhanced tumor growth, we depleted CD4$^+$ T cells from tumor-bearing mice. Interestingly, CD4$^+$ T cell depletion resulted in reduced tumor growth in both Asm-WT and Asm-KO mice. This is in line with studies by Ueha et al., who observed a reduction of tumor growth after CD4$^+$ T cell depletion (*Ueha et al., 2015*). However, the tumor growth in Asm-deficient mice was still significantly enhanced compared to WT littermates after CD4$^+$ T cell depletion. Subsequent analysis of CD8$^+$ T cells revealed reduced activation of CD8$^+$ T cells from tumor-bearing Asm-deficient mice. These results provide evidence that Asm activity has an impact not only on CD4$^+$ T cell subsets but also on CD8$^+$ T cell function during tumorigenesis. A reduced capacity to release cytotoxic granules by antigen-specific CD8$^+$ T cells from LCMV-infected Asm-deficient mice has already been described by *Herz et al., 2009*. In accordance, we also observed decreased frequencies of granzyme B producing CD8$^+$ TILs in tumor-bearing Asm-deficient mice, after CD4$^+$ T cell depletion. Further analysis of CD8$^+$ T cells from T cell-specific Asm-deficient mice (Asm/CD4cre) revealed a reduced activation upon stimulation in vitro compared to WT controls. Strikingly, Asm deficiency was associated with a lowered in vitro killing capacity of CD8$^+$ CTLs, suggesting that

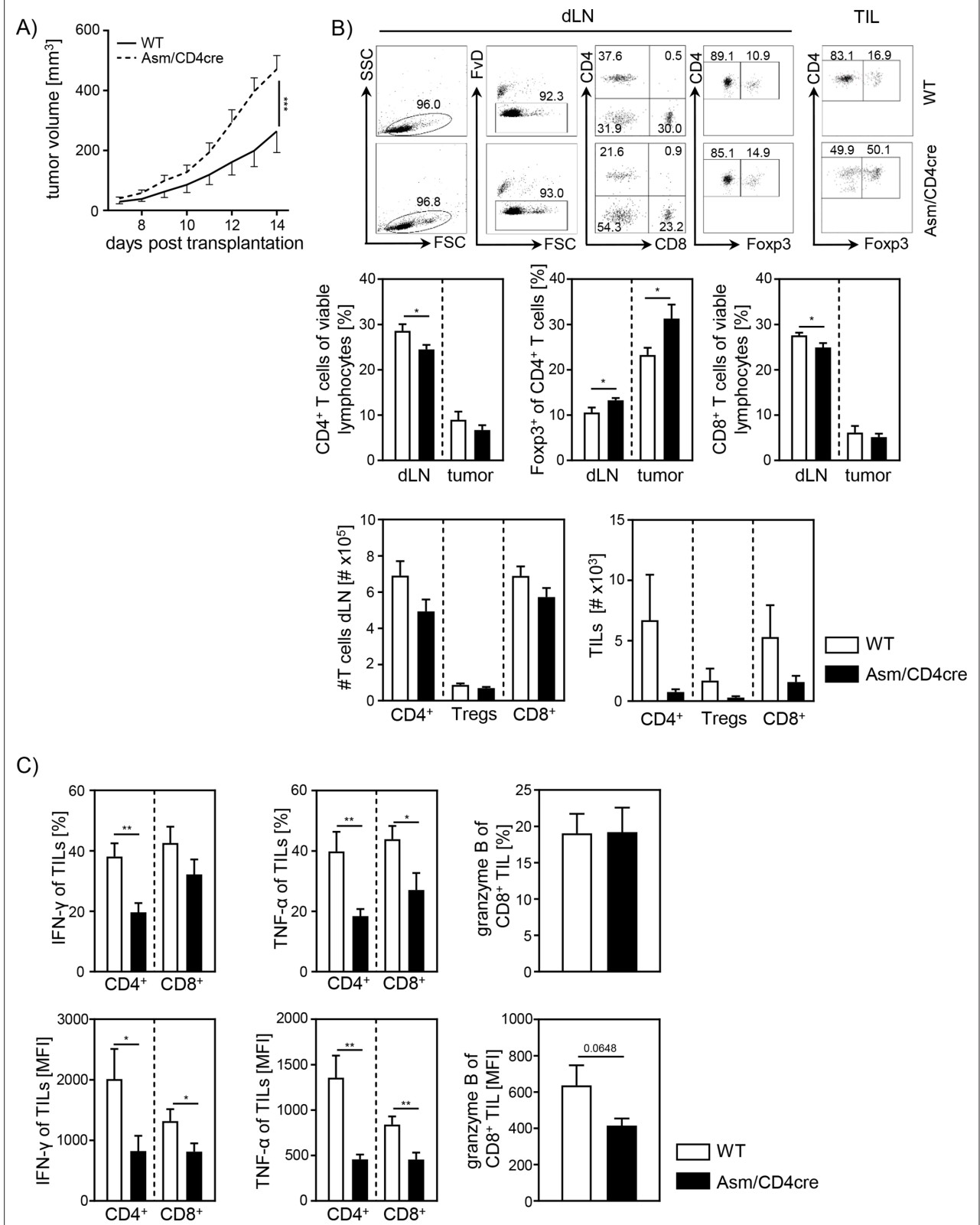

**Figure 4.** Cell-intrinsic acid sphingomyelinase (Asm) activity determines CD8+ T cell activation in vivo. (**A**) B16-F1 melanoma cells were transplanted into Asm/CD4cre mice and wildtype (WT) littermates, and tumor growth was monitored when tumors reached a detectable size (n=12–16). (**B**) Percentages of CD4+ T cells, Foxp3+ Tregs, and CD8+ T cells within dLN and tumor were determined by flow cytometry, and absolute cell numbers were calculated. Representative dot plots are shown in the upper panel. (**C**) Expression of IFN-γ, TNF-α, and granzyme B of tumor-infiltrating lymphocytes (TILs) was

*Figure 4 continued on next page*

*Figure 4 continued*
determined by flow cytometry. Results from four independent experiments are depicted as mean ± SEM. Statistical analysis was performed by two-way ANOVA with Sidak's multiple comparisons, Mann-Whitney U-test, or Student's t-test. (*p<0.05, **p<0.01, ***p<0.001).

The online version of this article includes the following source data for figure 4:

**Source data 1.** Cell-intrinsic Asm activity determines CD8+ T cell activation in vivo.

the ceramide content could regulate cytotoxic CD8+ T cell function. Indeed, Asm-deficient CD8+ T cells with reduced ceramide concentrations showed an impaired granzyme B production. Interestingly, this phenotype could partially be rescued by the addition of C16 ceramide in vitro. Validating the impact of ceramide on CD8+ T cell responses, we used Ac/CD4cre mice with T cell-specific ablation of Ac expression resulting in increased ceramide concentrations in CD8+ T cells. Well in line with results from Asm-deficient CD8+ T cells, we demonstrated, at least to our knowledge, for the first time, that Ac-deficient CD8+ T cells show increased granzyme B expression upon stimulation and importantly, elevated in vitro killing activity. This phenotype correlated with a facilitated anti-tumoral T cell response leading to reduced tumor growth rates in Ac/CD4cre mice transplanted with B16-F1 melanoma cells in contrast to WT littermates. These results indicate that ceramide concentrations may determine T cell activity and function in vitro and in vivo. In previous studies, intracellular S-1-P has been shown to reduce anti-tumor functions of T cells (*Chakraborty et al., 2019*; *Olesch et al., 2020*). To exclude that the modulated anti-tumoral T cell response in Asm/CD4cre and Ac/CD4cre mice, respectively, is caused by altered S-1-P concentrations, we quantified its abundance in CD3/CD28 stimulated T cells. However, S-1-P concentrations were under the detection limit of 0.5 pmol per 1×10⁶ cells (data not shown), emphasizing that indeed ceramide is more likely to modulate the anti-tumoral T cell response than S-1-P levels.

Our results demonstrate that ablation of Ac in CD8+ T cells from Ac/CD4cre mice led to a facilitated phosphorylation of ZAP-70 and PLCγ suggesting that ceramide, most likely as ceramide-enriched platforms in the plasma membrane, modulates the TCR signaling pathway. In accordance Bai and colleagues observed reduced phosphorylation of different TCR signaling molecules in stimulated human CD4+ T cells in the presence of the Asm inhibitor imipramine (*Bai et al., 2015*), further

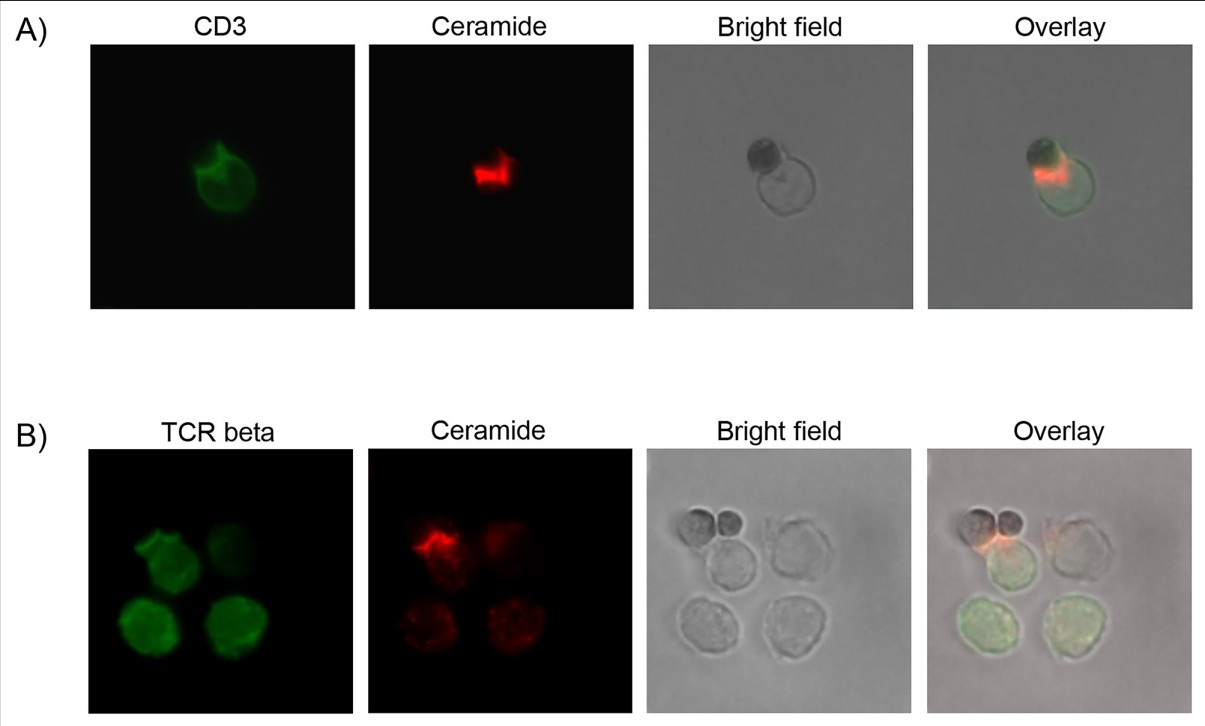

**Figure 5.** Ceramide co-localizes with CD3 and T cell receptor (TCR). CD8+ T cells were isolated and stimulated with CD3/CD28 MACSiBead particles for 2 hr and stained for ceramide (red) and (**A**) CD3 or (**B**) TCR beta (green). Cells were visualized using a Biorevo BZ-9000 fluorescence microscope.

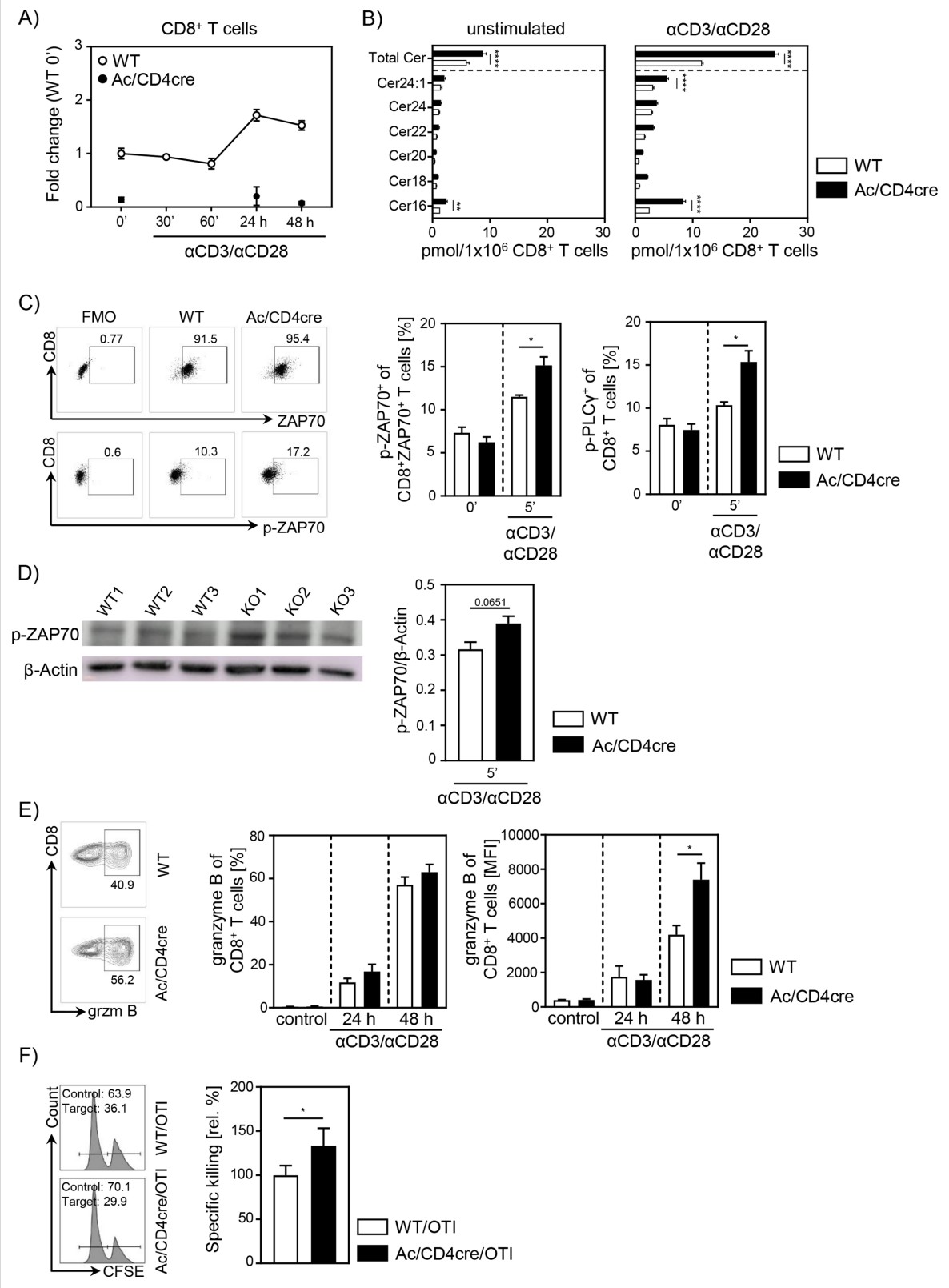

**Figure 6.** Acid ceramidase (Ac)-deficient CD8+ T cells have elevated ceramide levels and show increased activation in vitro. (**A**) Isolated CD8+ T cells from *Asah1*fl/fl/*Cd4*cre/+ mice (Ac/CD4cre) and *Asah1*fl/fl/*Cd4*+/+ littermates (wildtype [WT]) where either left unstimulated or stimulated with anti-CD3 and anti-CD28 for indicated time points. mRNA expression of Ac (*Asah1*) following activation was analyzed by RT-qPCR (n=3–4). (**B**) Ceramide levels of CD8+ T cells were determined by mass spectrometry (n=4). (**C**) For T cell receptor signaling analysis, splenocytes from Ac/CD4cre and WT mice were left

*Figure 6 continued on next page*

*Figure 6 continued*

unstimulated (0') or stimulated with anti-CD3 and anti-CD28 for 5 (5') min. Afterward, samples were analyzed for phospho-ZAP70 of gated ZAP70+CD8+ T cells and phospho-PLCγ of gated CD8+ T cells by flow cytometry (n=4). Representative dot plots and fluorescence minus one (FMOs) for phospho-ZAP70 are shown in the left panel. (**D**) Western blot analysis of phospho-ZAP70 expression of CD8+ T cells from Ac/CD4cre and WT mice after 5 min of stimulation with anti-CD3 and anti-CD28 (n=3). (**E**) CD8+ T cells were left untreated as control or stimulated for 24 or 48 hr and analyzed for granzyme B expression by flow cytometry (n=5–8). Representative contour plots are shown in the left panel. (**F**) Specific killing of antigen-specific cytotoxic lymphocytes from Ac/CD4cre/OTI mice and WT controls was assessed (n=8–9). Representative histograms are shown in the left panel. Data are depicted as mean ± SEM. Statistical analysis was performed by two-way ANOVA with Sidak's multiple comparisons, Mann-Whitney U-test, or Student's t-test. (*p<0.05, **p<0.01, ****p<0.0001).

The online version of this article includes the following source data and figure supplement(s) for figure 6:

**Source data 1.** Ac-deficient CD8+ T cells have elevated ceramide levels and show increased activation in vitro.

**Figure supplement 1.** In vitro characterization of acid ceramidase (Ac)-deficient CD4+ T cells.

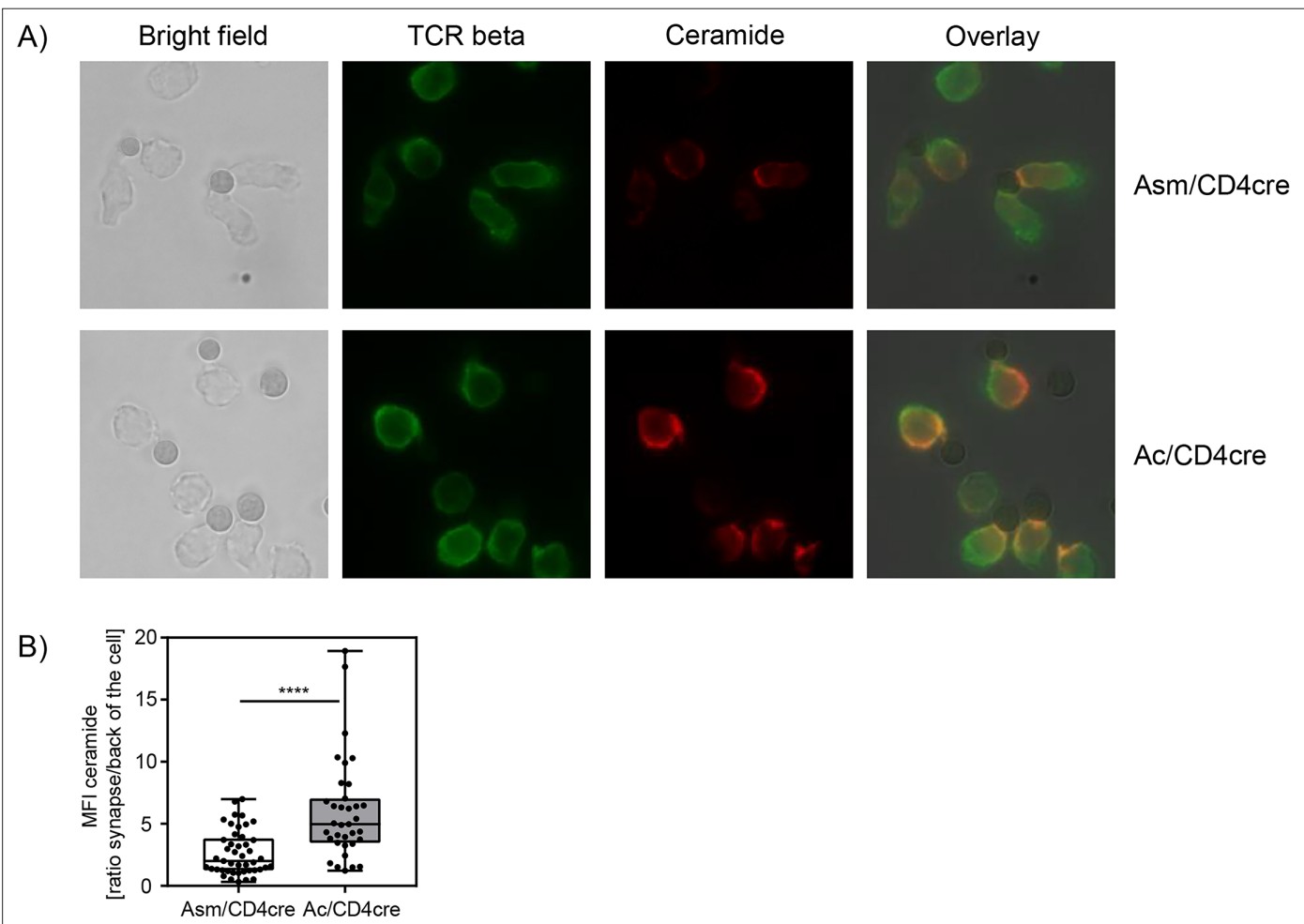

**Figure 7.** Synaptic ceramide is elevated in acid ceramidase (Ac)-deficient T cells compared to acid sphingomyelinase (Asm)-deficient T cells. (**A**) CD8+ T cells were isolated from Asm/CD4cre or Ac/CD4cre mice and stimulated with CD3/CD28 MACSiBead particles for 2 hr and stained for ceramide (red) and T cell receptor beta (green). Cells were visualized using a Biorevo BZ-9000 fluorescence microscope. (**B**) The ceramide signal was quantified in the synapse and in the back of cell, and the ratio of the signals was calculated (n=35–45). Data are depicted as boxplot with min/max whiskers. Statistical analysis was performed by Student's t-test. (****p<0.0001).

The online version of this article includes the following source data for figure 7:

**Source data 1.** Synaptic ceramide is elevated in Ac-deficient T cells compared to Asm-deficient T cells.

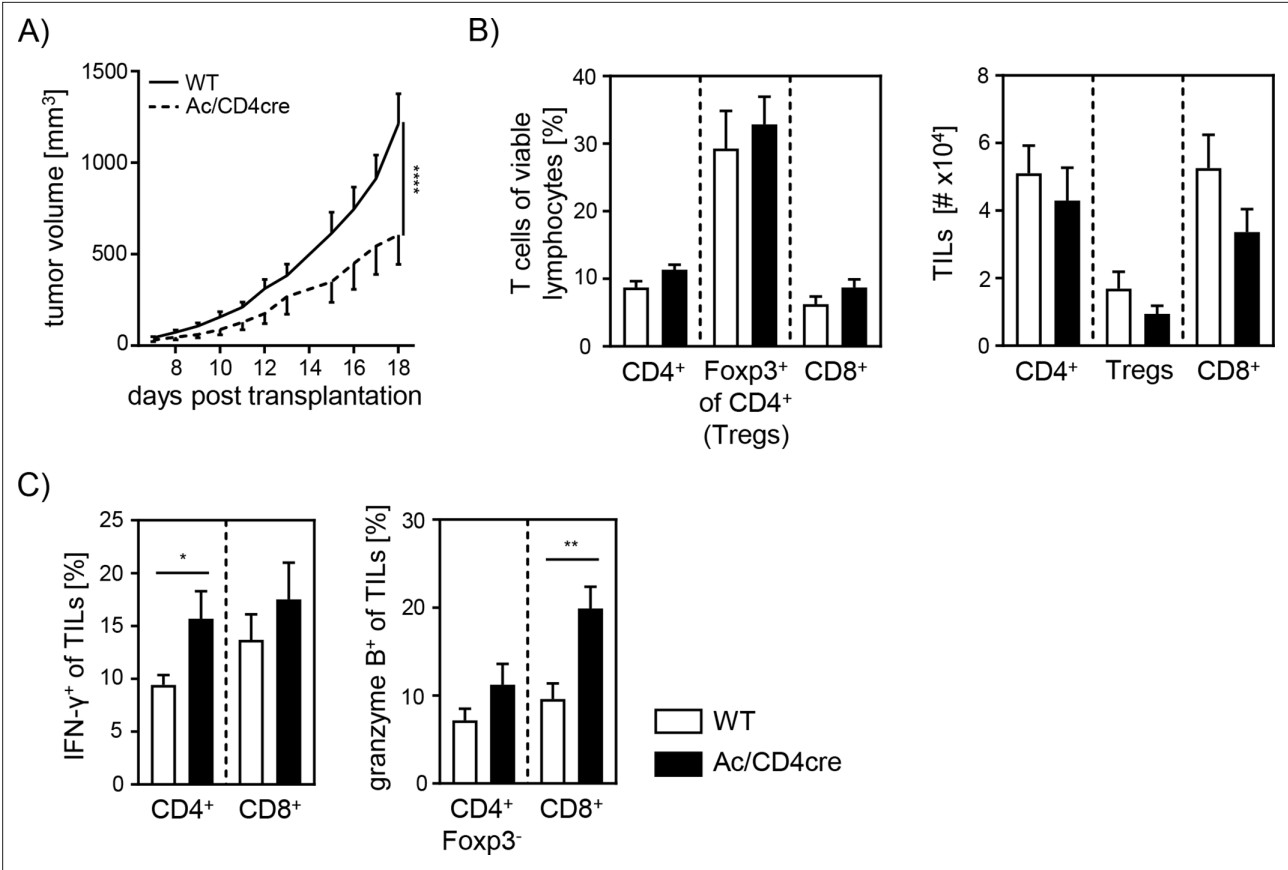

**Figure 8.** Elevated anti-tumor immune response in T cell-specific acid ceramidase (Ac)-deficient mice. (**A**) B16-F1 melanoma cells were transplanted into Ac/CD4cre mice and wildtype (WT) littermates. Tumor volume was monitored once tumors have been established (n=8–10). (**B**) Percentages of CD4+ T cells, Foxp3+ Tregs, and CD8+ T cells were determined by flow cytometry, and absolute cell numbers were calculated. (**C**) Frequencies of IFN-γ+ and granzyme B+ tumor-infiltrating lymphocytes (TILs) were determined by flow cytometry. Results from two independent experiments are depicted as mean ± SEM. Statistical analysis was performed by two-way ANOVA with Sidak's multiple comparisons or Student's t-test. (*p<0.05, **p<0.01, ****p<0.0001).

The online version of this article includes the following source data for figure 8:

**Source data 1.** *Figure 8* Elevated anti-tumor immune response in T cell-specific Ac-deficient mice.

suggesting that TCR signaling is affected by ceramide. In addition, one might speculate about a positive feedback regulation between ceramide and T cell activation since TCR stimulation of CD4+ as well as CD8+ T cells resulted in elevated ceramide concentration, which in turn seems to strengthen the TCR signaling cascade.

By using fluorescence microscopy, we were able to reveal a co-localization of ceramide and TCR in stimulated CD8+ T cells. Strikingly, synaptic ceramide in CD8+ T cells from Ac/CD4cre mice was highly elevated compared to that of Asm-deficient CD8+ T cells. These results indicate that Asm and Ac activity in T cells not only effect the overall cellular ceramide content but also ceramide levels within the immunological synapse, supporting our hypothesis that the ceramide level is involved in the strength of TCR-induced signaling pathway. Although our results clearly demonstrated the impact of Asm and Ac activity on the ceramide content and T cell function, we could not exclude that other enzymes of the sphingolipid pathways may contribute to increased or decreased ceramide levels. For instance, ceramide could also be synthesized by the neutral sphingomyelinase 2, which was proposed to be required for the immune synapse polarization of TCR signaling components in human Jurkat T cells (*Börtlein et al., 2018*). Interestingly, blocking its enzymatic activity interferes with the polarized release of exosomes from multivesicular bodies produced by T cell clones (*Mittelbrunn et al., 2011*), emphasizing an important and diverse role of synaptic ceramide in T cell function. However, ablation of Asm resulted in lower cellular and synaptic ceramide levels associated with impaired CD8+ T cell

function, suggesting that other sphingomyelinases are not able to fully compensate for Asm deficiency in T cells.

Overall, our study provides evidence that the cell-intrinsic ceramide content is regulated by the activity of Asm and Ac and associated with CD4[+] as well as CD8[+] T cell function in vitro and in vivo. Thereby, the sphingolipid metabolism represents a potential therapeutic target for improving antitumoral T cell responses during tumorigenesis. However, the use of drugs modulating ceramide generating enzymes like amitriptyline or other FIASMAs should be carefully reflected in cancer patients. Nevertheless, genetic engineering of T cells by manipulating the expression of sphingolipid metabolizing enzymes and using them for cancer therapy could be a potential approach for cancer therapy.

## Materials and methods

### Mice

All mice were on C57BL/6 background and maintained under specific pathogen-free conditions at the Animal Facility of University Hospital Essen. Female C57BL/6 mice were purchased from Envigo Laboratories (Envigo CRS GmbH, Rossdorf, Germany).

Asm-KO (*Smpd1tm1Esc*) mice has been described previously (*Horinouchi et al., 1995*). Floxed *Smpd1* (*Smpd1tm1a(EUCOMM)Wtsi*) and *Asah1* (*Asah1tm1.1Jhkh*) mice (*Gulbins et al., 2013*) were crossed to *Cd4cre* mice and for some experiments additionally to OTI mice (*Hogquist et al., 1994*) expressing a transgenic TCR recognizing ovalbumin peptide 257–264 (kindly provided by Tetyana Yevsa, Hannover Medical School, Germany). All animal experiments were carried out in accordance with the guidelines of the German Animal Protection Law and were approved by the state authority for nature, environment, and customer protection, North Rhine-Westphalia, Germany.

### Cell lines

B16-F1 (CRL-6323) melanoma cells were cultured in Iscove's Modified Dulbecco's Medium(IMDM) supplemented with 10% heat-inactivated fetal calf serum (FCS), 25 µM β-mercaptoethanol, and antibiotics [100 U/ml penicillin, 0.1 mg/ml streptomycin](IMDM complete). Cells were maintained in a humidified 5% $CO_2$ atmosphere at 37°C. Cells were stored in liquid nitrogen and passaged twice before transplantation. Mycoplasma testing was performed every other month by PCR in in vitro propagated cultures.

### Tumor transplantation

Tumor cells were harvested and washed twice with PBS. $5×10^5$ tumor cells in a volume of 100-µl PBS were injected subcutaneously (s.c.) into the right flank of experimental animals. Tumor volume was calculated using the formula V = (W$^2$ × L)/2 (*Faustino-Rocha et al., 2013*) based on caliper measurements once tumors have established.

### Amitriptyline treatment

B16-F1 melanoma cells were injected s.c. into 8–12-week-old C57BL/6 mice, following mice received 20-mg amitriptyline/kg bodyweight in 100-µl PBS via daily i.p. injection over a period of 13 days.

### Cell isolation and activation

Single-cell suspensions of splenocytes were generated by rinsing spleens with erythrocyte lysis buffer and washing with PBS supplemented with 2% FCS and 2 mM EDTA. T cells were isolated from splenocytes either by using the CD4[+] or CD8[+] T cell isolation kit (Miltenyi Biotec, Bergisch Gladbach, Germany) according to the manufacturer's recommendation alone or followed by anti-CD4, anti-CD25, anti-CD8 staining, and cell sorting using an Aria II Cell Sorter (BD Biosciences, Heidelberg, Germany). T cells were stimulated with 1 µg/ml anti-CD3 plate-bound and 1 µg/ml anti-CD28 soluble (both BD Biosciences, Heidelberg, Germany) in IMDM complete culture medium. For exogenous ceramide administration in vitro C16 ceramide (Avanti Polar Lipids, Birmingham, USA) solved in 100% EtOH was sonicated for 10 min. The final concentration used for in vitro T cell culture was 5 µM.

## Cell isolation from dLN and tumors

dLN were pestled through a 70-μm cell strainer and washed with PBS containing 2-mM EDTA and 2% FCS. Tumors were homogenized and pestled through a 70-μm cell strainer and washed with IMDM complete culture medium.

## T cell differentiation

For Treg differentiation (iTreg), CD4$^+$CD25$^-$ T cells were stimulated with anti-CD3/anti-CD28 as described above in the presence of 20 ng/ml IL-2 (eBioscience, ThermoFisher Scientific, Langenselbold, Germany) and 5 ng/ml TGF-β1 (R&D Systems, Bio-Techne, Wiesbaden, Germany) for 72 hr.

## Killing assay

For the generation of antigen-specific CTLs, splenic CD8$^+$ T cells from Asm/CD4cre/OT-I, Ac/CD4cre/OT-I mice or the respective littermate controls were cultivated in the presence of irradiated splenocytes, 1 μg/ml OVA-peptide 257–264, 10 ng/ml IL-2, and 20 ng/ml IL-12 for 6 days. As control, cells were stimulated without OVA-peptide 257–264 (non-CTLs). At day 3, cells were split, and fresh IMDM complete culture medium supplemented with 10 ng/ml IL-2 was added. CTLs and non-CTLs were isolated from the culture and incubated with OVA-peptide 257–264 loaded Carboxyfluoresceinsuccinimidylester$^{high}$(CFSE$^{high}$)-labeled (2.5 μM CFSE) target and unloaded CFSE$^{low}$-labeled (0.25 μM CFSE) control cells (both splenocytes from WT mice) for 2 or 4 hr. Frequencies of target and control populations were analyzed using flow cytometry. Specific killing was calculated as described before (*Barber et al., 2003*) using the formula: specific killing [%]=100−([(CTL$^{target}$/CTL$^{control}$)/(non-CTL$^{target}$/non-CTL$^{control}$)] × 100).

## Antibodies and flow cytometry

Anti-CD4, anti-CD8, anti-IFN-γ, anti-CD25, anti-CD44 (BD Biosciences, Heidelberg Germany), anti-CD8, anti-Foxp3, anti-TNF-α, anti-CD69 (eBioscience, ThermoFisher Scientific, Langenselbold, Germany), anti-p-PLCγ, anti-p-ZAP70 (Cell Signaling, Frankfurt am Main, Germany), anti-Granzyme B (Invitrogen, ThermoFisher Scientific, Langenselbold, Germany), and anti-ZAP70 (BioLegend, San Diego, USA) were used as fluorescein isothiocyanate (FITC), pacific blue, phycoerythrin (PE), allophycocyanin, AlexaFluor488, AlexaFluor647, PE-cyanin 7, or peridinin-chlorophyll protein conjugates. Dead cells were identified by staining with the fixable viability dye eFluor 780 (eBioscience, ThermoFisher Scientific, Langenselbold, Germany). Intracellular staining for Foxp3 and Granzyme B was performed with the Foxp3 staining kit (eBiocience, ThermoFisher Scientific, Langenselbold, Germany) according to the manufacturer's protocol. IFN-γ and TNF-α expression were measured by stimulating cells with 10 ng/ml phorbol 12-myristate 13-acetate and 100-μg/ml ionomycin (both Sigma-Aldrich, München, Germany) for 4 hr in the presence of 5-μg/ml Brefeldin A (Sigma-Aldrich, München. Germany), followed by treatment with 2% paraformaldehyde and 0.1% IGEPAL CA-630 (Sigma-Aldrich, München, Germany), and staining with the respective antibody for 30 min at 4°C. Flow cytometric analyses were performed with an LSR II and a Canto II instrument using DIVA software (BD Biosciences, Heidelberg Germany).

## Ceramide and S-1-P quantification by HPLC-MS/MS

Cell suspensions were subjected to lipid extraction using 1.5-ml methanol/chloroform (2:1, v:v) as described (*Gulbins et al., 2018*). The extraction solvent contained C17 ceramide (C17 Cer) and d$_7$-S-1-P (both Avanti Polar Lipids, Alabaster, USA) as internal standards. Chromatographic separations were achieved on a 1290 Infinity II HPLC (Agilent Technologies, Waldbronn, Germany) equipped with a Poroshell 120 EC-C8 column (3.0×150 mm, 2.7 μm; Agilent Technologies). MS/MS analyses were carried out using a 6495 triple-quadrupole mass spectrometer (Agilent Technologies) operating in the positive electrospray ionization mode (ESI+) (*Naser et al., 2020*). The following mass transitions were recorded (qualifier product ions in parentheses): $m/z$ 380.3 → 264.3 (82.1) for S-1-P, $m/z$ 387.3 → 271.3 (82.1) for d$_7$-S-1-P, $m/z$ 520.5 → 264.3 (282.3) for C16 Cer, $m/z$ 534.5 → 264.3 (282.3) for C17 Cer, $m/z$ 548.5 → 264.3 (282.3) for C18 Cer, $m/z$ 576.6 → 264.3 (282.3) for C20 Cer, $m/z$ 604.6 → 264.3 (282.3) for C22 Cer, $m/z$ 630.6 → 264.3 (282.3) for C24:1 Cer, and $m/z$ 632.6 → 264.3 (282.3) for C24 Cer. Peak areas of Cer subspecies, as determined with MassHunter software (Agilent Technologies), were normalized to those of the internal standard (C17 Cer) followed by external calibration in the range

of 1 fmol–50 pmol on column. S-1-P was directly quantified via its deuterated internal standard $d_7$-S-1-P (0.125 pmol on column). Determined ceramide and S-1-P amounts were normalized to cell count.

## T cell receptor signaling

For analyzing phosphorylation of TCR signaling molecules by flow cytometry, $5×10^5$ splenocytes were left unstimulated or stimulated with 1 µg/ml anti-CD3 and 1 µg/ml anti-CD28 for 5 or 10 min, treated with Cytofix/Cytoperm (BD Biosciences, Heidelberg Germany) for 1 hr, and stained with the respective antibodies for 30 min at 4°C. For western blot analysis isolated T cells were stimulated for 5 min, washed with PBS, collected in lysis buffer, and incubated on ice for 20 min. Afterward, cells were centrifuged for 5 min at 1200 rpm, the supernatant was collected, and protein concentrations were determined as described by *Lowry and Randall, 1951*. 30 µg of total protein were diluted in sodium dodecyl sulfate (SDS)-buffer, denatured at 95 °C for 5 min and subjected to SDS polyacrylamide gel electrophoresis. Separated proteins were transferred to a Polyvinylidendifluorid(PVDF) membrane using the Trans-Blot Turbo RTA Transfer Kit (Bio-Rad Laboratories, Feldkirchen, Germany) according to the manufacturer's recommendations. The PVDF membrane was blocked with 5% BSA in TBS-T for 1 hr at room temperature. Primary antibodies against p-ZAP70 (Cell Signaling, Frankfurt am Main, Germany, 1:1000) and β-actin (Sigma-Aldrich, Saint Louis, USA, 1:2000) were incubated over night at 4°C. Secondary anti-rabbit IgG antibody (Sigma-Aldrich, Saint Louis, USA, 1:10,000) was incubated for 1 hr at room temperature. Blots were developed using SuperSignal West Femto Maximum Sensitivity Substrate (Thermo Scientific), and signals were detected with a Fusion FX System (Vilber, Eberhardzell, Germany).

## Microscopy

Isolated CD8[+] T cells were plated in ibidi µ-slides (Ibidi, Gräfelfing, Germany) and stimulated with CD3/CD28 MACSiBead particles (Miltenyi Biotec, Bergisch Gladbach, Germany) in a 1:1 ratio for 2 hr at room temperature. Afterward, cells were fixed with 4% PFA, blocked with 1% BSA in PBS for 30 min, and stained with anti-ceramide antibody (LSBio, Seattle, USA, 1:20 in 1% BSA in PBS) for 1 hr. Secondary anti-mouse IgM antibody (BioLegend, San Diego, USA, PE-conjugated, 1:1000) and anti-CD3 (BioLegend, San Diego, USA, FITC-conjugated) or anti-TCR beta (Invitrogen, Carlsbad, USA, FITC-conjugated, 1:50) antibodies were diluted in 1% BSA in PBS and incubated for 30 min at room temperature. Stained cells were mounted using fluorescence mounting medium (Dako, California, USA) and visualized with a Biorevo BZ-9000 fluorescence microscope (Keyence, Itasca Illinois, USA). For quantification, the ceramide signal was measured in the synapse and in the back of the cell and calculated using following formula: *Fluorescence intensity = Integrated Density – (Area of Selected Cell × Mean Fluorescence of Background readings)*.

## CD4[+] T cell depletion

For depletion of CD4[+] T cells during tumorigenesis, 200-µg anti-mouse CD4-depleting antibody (clone GK1.5; BioXcell, Lebanon, USA) was injected intraperitoneal on days –1, 3, 6, 9, and 12 after tumor cell transplantation.

## RNA isolation, cDNA synthesis, and qRT-PCR

RNA was isolated using the NucleoSpin RNA XS Kit (Macherey-Nagel, Düren, Germany) according to the manufacturer's instructions. 100 ng of RNA was reversed transcribed using M-MLV reverse transcriptase (Promega, Mannheim, Germany) with dNTPs (Bio-Budget, Krefeld, Germany), Oligo-dT mixed with Random Hexamer primers (both Invitrogen, Frederick Maryland, USA). Quantitative real-time PCR was performed using the Fast SYBR Green Master Mix (Thermo Fisher Scientific, Braunschweig, Germany) and a 7500 Fast Real-Time PCR System (Thermo Fisher Scientific, Darmstadt, Germany). Samples were measured as technical duplicates. Expression levels were normalized against ribosomal protein S9 (RPS9). Following primer sequences were used: Asm (*Smpd1*) CTG TCA GCC GTG TCC TCT TCC TTA, GGG CCC AGT CCT TTC AAC AG, Ac (*Asah1*) TTC TCA CCT GGG TCC TAG CC, TAT GGT GTG CCA CGG AAC TG, RPS9 CTG GAC GAG GGC AAG ATG AAG C, TGA CGT TGG CGG ATG AGC ACA.

## Statistical analysis

Statistical analyses were calculated using Graph Pad Prism Software (Graph Pad Software, La Jolla, CA). To test for Gaussian distribution, D'Agostino-Pearson omnibus and Shapiro-Wilk normality tests

were used. If data passed normality testing, paired or unpaired Student's t-test was performed, otherwise Mann-Whitney U-test was used for unpaired data. Differences between two or more groups with different factors were calculated using two-way ANOVA followed by Sidak's post-test. Statistical significance was set at the levels of $*p<0.05$, $**p<0.01$, $***p<0.001$, and $****p<0.0001$.

## Acknowledgements

We kindly thank Sina Luppus for excellent technical assistance and Witold Bartosik and Christian Fehring for cell sorting. Moreover, we thank Daniel Herrmann for the help with the HPLC-MS analyses of ceramides. This work was supported by the Deutsche Forschungsgemeinschaft (DFG - GRK2098 to AMW, JB, KAB, EG and WH, and GRK1949 to AMW, JB, and WH).

## Additional information

### Funding

| Funder | Grant reference number | Author |
| --- | --- | --- |
| Deutsche Forschungsgemeinschaft | GRK1949 | Astrid M Westendorf |
| Deutsche Forschungsgemeinschaft | GRK2098 | Katrin Anne Becker |

The funders had no role in study design, data collection and interpretation, or the decision to submit the work for publication.

### Author contributions

Matthias Hose, Anne Günther, Conceptualization, Data curation, Investigation, Visualization, Methodology, Writing – original draft, Writing – review and editing; Eyad Naser, Tina Schönberger, Julia Falkenstein, Athanasios Papadamakis, Methodology; Fabian Schumacher, Erich Gulbins, Investigation, Methodology; Burkhard Kleuser, Supervision; Katrin Anne Becker, Investigation; Adriana Haimovitz-Friedman, Conceptualization; Jan Buer, Astrid M Westendorf, Conceptualization, Funding acquisition; Wiebke Hansen, Conceptualization, Resources, Data curation, Formal analysis, Supervision, Funding acquisition, Visualization, Writing – original draft, Project administration, Writing – review and editing

### Author ORCIDs

Matthias Hose http://orcid.org/0000-0003-0746-5591
Fabian Schumacher http://orcid.org/0000-0001-8703-3275
Athanasios Papadamakis http://orcid.org/0000-0002-4095-4242
Adriana Haimovitz-Friedman http://orcid.org/0000-0002-4884-0581
Jan Buer http://orcid.org/0000-0002-7602-1698
Astrid M Westendorf http://orcid.org/0000-0002-2121-2892
Wiebke Hansen http://orcid.org/0000-0002-6020-0886

### Ethics

All experiments were performed in strict accordance with the guidelines of the German Animal Protection Law and approved by the State Agency for Nature, Environment, and Consumer Protection (LANUV), North Rhine-Westphalia, Germany (Az 84-02.04.2015.A367, Az 84-02.04.2016.A506, Az 84-02.04.2017.A024).

### Decision letter and Author response

Decision letter https://doi.org/10.7554/eLife.83073.sa1
Author response https://doi.org/10.7554/eLife.83073.sa2

## Additional files

### Supplementary files
• MDAR checklist

## Data availability

Source Data files have been provided for figures 1-8 and supplemental figures.

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
