## [Editor Report]

The authors have used mouse genetics to show that acid sphingomyelinase mediated generation of ceramide promotes T cell activation and tumour control, whereas acid ceramidase reduces ceramide levels and impairs T cell activation and tumour control. The results also show that ceramide is polarised toward the immunological synapse. The work will be of relevance to those studying the role of lipids in signaling reactions generally and specifically to fields of T cell activation and tumour immunology.

---

## [Decision Letter]

[Editors' note: this paper was reviewed by Review Commons.]

---

## [Author Response]

Reviewer #1:Major Comments:1) There needs to be a quantification of S1P in T cells in response to ASM or AC knockouts in T cells in Figures2A and 6A, as intracellular S1P has been shown to reduce anti-tumor functions of T cells in previous studies.

We thank the reviewer for this helpful comment. Indeed, we have now analysed S1P levels in Asm- and Ac-deficient T cells. However, S1P concentrations were under detection limit and therefore not included in the revised manuscript, but we included the information about non-detectable S1P levels in the Discussion section of the revised manuscript

2) The tumor volume measurements were performed within 13-18 days of post-injection of cells in mice. These studies are too short and need to be evaluated at around 21-30 days to evaluate whether observed defects in T cells are sustained in a time-dependent fashion.

We agree that analysis of our experiments at later time points would be interesting to see how T cell responses develop over time. Therefore, we repeated tumor transplantation into Ac^flox/flox^/CD4cre mice to evaluate tumor growth and T cell responses over an extended period of time. However, for animal welfare reasons, we again had to sacrifice the mice on day 18 after transplantation since tumors have become too large.

3) The mechanisms that are involved in the regulation of anti-tumor functions of T cells by ASM versus AC are not very clear. Although TCR signaling is implicated, the mechanistic aspects of the data are weak.

We hypothesize that the clustering of ceramide facilitates the polarization of TCR molecules and thereby foster the intracellular downstream signalling cascade. Indeed, microscopy of T cells revealed a co-localization of TCR β and ceramide upon stimulation with CD3/CD28 MACSiBeads. We have included these new data in Figure 5 of the revised manuscript. Moreover, we could confirm and have included new data on a facilitated phosphorylation of Zap70 in ceramide enriched T cells from Ac^flox/flox^/CD4cre mice analysed by Western Blot (Figure 6D of the revised manuscript), suggesting that an increase in ceramide-enriched platforms contribute to elevated TCR clustering and signalling

4) The subcellular localization of ceramide in the plasma membrane/TCR synapsis needs to be examined and shown to strengthen the major point of the study.

We thank the reviewer for this helpful suggestion. We performed fluorescence microscopy of CD8^+^ T cells stimulated with CD3/CD28 MACSiBead Particles and stained for CD3, TCR β and ceramide. Indeed, ceramide accumulates at the contact site with the particle and co-localizes with CD3 and the TCR. We included representative images in Figure 5 of the revised manuscript.

Minor comments:1) The effects of ASM or AC deletions on TCR signaling in T cells should be shown by Western blotting and/or immunofluorescence in addition to the flow cytometry-based assays (for detecting p-AKT, pZAP70, and p-PLC (these need to be examined in Figure 2F as in 6B). PKC-theta should be added in this panel also.

Analysis of phospho-TCR signalling molecules by flow cytometry is a well-established method and has been already been published by us and many other authors (for example, Hose et al., Front Immunol 2019, 10: 1225; Morelli et al., Cell Reports 2020, 30: 3448-65; Hong et al., Cell 2020, 180: 847-61; Balyan et al., Immunology 2018, 156: 384-401). Moreover, reduced phosphorylation of TCR signalling molecules upon Asm inhibition has already been shown by Western Blot analysis in a previous study (Bai, et al., Cell Death Dis. 2015,6(7):e1828.). Nevertheless, as a proof of concept we confirmed our results on elevated p-ZAP70 in Ac-proficient CD8^+^ T cells by western blot analysis as an example and included it in Figure 6 of the revised manuscript.

2) The list of references contain too many review articles, and it seems that it is missing many recent publications describing the roles and mechanisms of ceramide or S1P in the regulation of anti-tumor functions of T cells.

We thank the reviewer for this recommendation and have now included more primary and recent publications on the topic in the revised manuscript.

Reviewer #2:Major Comments:Figure 1: Data inconsistency: The tumor growth in control mice varies considerably between experiments (e.g. Figure 1a WT d13 700mm3 vs Figure 1d and WT/PBS d13 200mm3) why is this the case? While the impact of tumor control in Asm deficient mice is overall convincing the analysis of the functional defect of Asm deficient T cells is superficial. Analysis of additional cytokines (TNF, IL2) would be important. IFN needs be depicted as % positive rather than MFI and representative FACS plots would further strengthen the data. Same applies to Ki67 and CD44.

We agree that tumor growth shows variation between the different experiments. One reason for that might be genetic differences between the WT controls used in the experiments. In Figure 1A WT mice are non-transgenic littermates from our in-house AsmKO breeding (AsmKO heterozygous x AsmKO heterozygous), whereas commercial available C57BL/6JOlaHsd mice from Envigo were used for amitriptyline treatment in Figure 1 of the original manuscript, now shown in the Supplemental Figure 1. Data for IFNγ and CD44 are now depicted as % in Figure 1 of the revised manuscript. Examples for representative FACS plots on IFNγ and CD44 expressing TILs are included in Figure 2 and an example for the gating strategy of CD4 and CD8 T cells used for all experiments is shown in Figure 4 of the revised manuscript.

Figure 2: The authors switch between Asm KO to Asm inhibition without providing further explanation (e.g. Figure 2E and F the only use the inhibition model). The authors should clarify this in the text. Could the phenotype of Asm deficiency be rescued by addition of ceramide species in vitro? This could establish a mechanistic link between ceramide abundance and T cell function. Asm and Ac expression should be shown on a protein level (if possible) or at least on an mRNA level over time following activation. (E) intracellular cytokine staining and representative FACS plots would strengthen the data. (F) This data is not convincing. How were pos cells gated given the minimal shift in staining?

We are grateful for these helpful comments. We agree that switching between Asm KO and Asm inhibition was confusing in the initial manuscript. Moreover, data on the impact of Asm inhibition on CD4 T cells in vitro has already been published by others (Bai et al., Cell Death Dis. 2015,6(7):e1828). Therefore, we have now excluded the in vitro data on Asm inhibition from the revised manuscript. In order to analyse whether the phenotype of Asm deficiency can be rescued by the addition of ceramide, we isolated CD8^+^ T cells from Asm^flox/flox^/CD4cre and WT mice and stimulated them in presence or absence of C16 ceramide and evaluated the expression of granzyme B. Indeed, addition of C16 ceramide partially rescued granzyme B expression of Asm-deficient CD8^+^ T cells. The data are included in Figure 3F of the revised manuscript. Using RT-qPCR, we analysed the expression of Asm and Ac in T cells on mRNA level. We included the data in the revised manuscript in Figure 3A for CD8^+^ T cells and Supplemental Figure S2 for CD4^+^ T cells from Asm^flox/flox^/CD4cre and WT mice and in Figure 6A for CD8^+^ T cells and Supplemental Figure S3 for CD4^+^ T cells from Ac^flox/flox^/CD4cre and WT mice. In addition, we now included the gating and FMOs for FACS analysis of TCR signalling in Figure 6C and Supplemental Figure S3 of the revised manuscript.

Figure 3: The authors should harmonize what they mean by "activation" among figures and use the same markers/molecules in every figure (e.g. IFN not shown in Figure 3, but shown in Figure 1, also applies to Figure 4,5 and 6). Otherwise, the comparison between different figures/conditions/models (e.g. CD4^+^ T cell depletion) is not feasible. (B) Absolute numbers of CD4 and CD8 T cells need to be shown. (c) Ki67 levels should be shown as % positive and supported by representative FACS plots.

We thank the reviewer for these recommendation. Unfortunately, we do not have analysed every marker/ molecule in all different models. However, since the main topic of our study is the impact of the sphingolipid metabolism on cytotoxic T cell function, we now mainly focused on IFNγ and granzyme B expression. Regarding absolute CD8^+^ T cell numbers, we agree with the reviewer and included respective data in Figure 2B of the revised manuscript. However, since CD4^+^ T cells are depleted, we could have only added absolute numbers of the control groups.

Figure 4: (B) Gzmb levels should be shown as % positive and supported by representative FACS plots. (C) The proliferation assay shows that the KO and WT cells proliferate similarly once they enter cell cycle. The main difference is that in Asm KO T cells a larger fraction does not enter cell cycle. It would strengthen the manuscript to elucidate why this is the case. The authors should temper their conclusions. Arguing that Asm is "essential" for T cell activation seems exaggerated based on the provided data.

We thank the reviewer for these helpful comments. In the revised manuscript we now show granzyme B expression as % and included representative FACS plot (Figure 3E). We agree with the reviewer that it would be interesting to elucidate why a larger fraction of AsmKO CD4^+^ T cells did not enter the cell cycle. However, the main focus of our study was the impact of the sphingolipid metabolism on T cell responses during tumorigenesis and its cytotoxic activity. Therefore, we have decided to exclude data on CD4^+^ T cell proliferation from the revised manuscript. We apologize for the exaggerated wording and have carefully adjusted corresponding statements in the revised manuscript.

Figure 5: (A) Absolute CD4, Treg cell and CD8 T cell numbers need to be shown to optimally interpret the data. Parental gating of the FACS plots should be shown/indicated. (B) Cytokine response and Gzmb expression should be shown as % positive rather than MFI. Based on the FACS plot the differences are likely more impressive than the MFIs suggest. (C) tumor growth WT control on d14 about 200mm3, this is not consistent with Figure 1a.

We agree with the reviewers suggestions and now included absolute numbers and the FACS gating strategy in Figure 4B in the revised manuscript. Moreover, we additionally show % of cytokine and granzyme B expression (Figure 4C). As mentioned above, differences in tumor growth curves are likely due to the use of different WT controls in the individual experiments. Data shown in Figure 1A are from non–transgenic WT littermates of our in-house AsmKO breeding (AsmKO heterozygous x AsmKO heterozygous), whereas WT controls in Figure 4 are littermates of Asm^flox/flox^CD4cre mice that do not express the cre recombinase.

Figure 6: Since the paper talks about T cell function in general and since this is a completely new mouse model (Acflox/flox/CD4Cre) and the first time this enzyme has been studied in T cells, the authors should show the phenotype and characterization of CD4^+^ T cells as well, and not only CD8^+^ T cells (Figures 6A-D). (B) This data is not convincing. How were pos cells gated given the minimal shift in staining? (C) Gzmb should be shown as % positive and representative FACS plots should be added. (E) Tumor growth curve at d18 does not match with the tumor size at day of sacrifize shown in the adjacent bar graph. Absolute numbers of CD4, Treg cells and CD8 T cells in tumors should be shown.

We are grateful for these helpful suggestions. We have now included data on Ac-deficient CD4^+^ T cells in Supplemental Figure S3 of the revised manuscript. As mentioned above, we have added the FACS gating and FMOs plots for the analysis of TCR signalling in the new Figure 6C. Moreover, we now additionally show % of granzyme B-expressing T cells and representative FACS plots in Figure 6E of the revised manuscript. We agree and apologize that the way the tumor size was presented for Ac^flox/flox^/CD4cre mice is misleading. Therefore, we have now only included data of mice sacrificed 18 days post transplantation and added absolute numbers of tumor-infiltrating T cells in Figure 7B in the revised manuscript.

Minor comments:Discussion line 496: "the sphingolipid metabolism represents a potential therapeutic target for improving anti-tumoral T cell responses during tumorigenesis". If would great if the authors speculated more about how this could be done; how could a target inhibition of T cell be achieved? Maybe engineering CD4^+^, CD8^+^ and Tregs by manipulating the expression of these sphingolipid metabolizing enzymes and using them for tumor therapy could be a potential approach.

We have now discussed this issue in more details in the revised manuscript.

Reviewer #3:Major Comments:1. Authors further argue that TCR signalling is affected in Asm KO and Ac KO cells. Data provided are not fully convincing. First, a strong stimulation by aCD3/aCD28 antibodies is used and in the most cases differences between WT and mutant cells is minor (Figure 2, Figure 6). Indeed, the use of antibodies with high affinity/avidity to the receptors may hide the real impact of changes on early TCR signalling events.

We agree with the reviewer that effects on phosphorylation of TCR signalling molecules seems not to be tremendous, but differences are significant after stimulation with aCD3/aCD28. Although differences in early TCR signalling events might be even stronger after a milder form of stimulation, we decided to use the same stimulation protocol for all of our analysis in the manuscript to make the data comparable.

2. Flow cytometry is used exclusively to show changes in phosphorylation of signalling molecules (ZAP70, Akt, PLCg, p38). Why ZAP70, Akt and PLCg are tested for Ac part and PLCg and p38 in the Asm section? Histograms are shown and the difference between control (WT, positive) and amitriptyline-treated T cells is very subtle. Negative control is not shown. In the graph, authors present % of cells with phosphorylated protein from the total number of protein-positive cells. This value strongly depends on setting of the value, which represents phoshoprotein positivity. For Ac mutant T cells, authors even use relative values in the graphs. Presented histograms again do not show well-distinguishable difference. However, it is unclear why authors use relative % values for data which represent ratiometric values. Presenting all measured and processed values in the Supplementary Data may help to resolve this issue. Still, the changes are small, and I am not sure if altered TCR signalling can explain well-documented effect of ceramide levels on T cell function in vivo and in vitro.

We totally agree with the reviewers’ opinion and apologize for the measurement of different signalling molecules for the Asm and Ac section. Since the section on Asm inhibition by amitriptyline was rather confusing (see comments of reviewer #2) and the impact of Asm inhibition on TCR signalling has already been shown in detail for CD4^+^ T cells by others (Bai et al., Cell Death Dis. 2015,6(7):e1828) we excluded these data from the revised manuscript. For Ac-deficient T cells we added the FACS gating and FMOs (Figure 6C and Supplemental Figure S3) and performed exemplary western blot analysis for p-ZAP70 (Figure 6D) to further strengthen these data. Moreover, we now present % instead of relative values in the revised manuscript.

3. Immunobloting can further support presented flow cytometry data related to TCR-induced signalling (phosphorylation of signalling molecules).

As mentioned above, we additionally performed western blot analysis to support the flow cytometry data.

4. As described above, the effect on early TCR signalling is inconclusive (in the current form of the manuscript). Please, revise the sentence in the Abstract: 'Mechanistically, our results indicate that ceramide levels, regulated by Asm and Ac activity, correlate with the phosphorylation of TCR signaling molecules in T cells.'It is not clear if authors mean that TCR signalling induces ceramide levels (well-supported by presented data) or that TCR signalling is affected by altered ceramide levels (less-well supported by the data).

We agree with the reviewer that our data provide evidence for an increase in ceramide after TCR activation. Nevertheless, unstimulated T cells deficient for Asm or Ac already showed differences in the ceramide levels (Figure 3B and 6B of the revised manuscript) and our new data provide evidence for co-localization of the TCR and ceramide upon stimulation (Figure 5 of the revised manuscript).

Therefore, one might speculate about an influence of ceramide levels on TCR signalling, which in turn further affects the ceramide levels. We have carefully rewritten this aspect in the Abstract and discussed this issue in the revised manuscript.

5. Ceramide levels: There is dramatic difference between values presented for ceramide in WT cells in Figure 2 and Figure 6. Please, comment on the difference. Adding all relevant numbers into a table would be very useful.

We agree that ceramide levels of WT cells shown in Figure 2 and Figure 6 of the original manuscript are different. However, ceramide concentration shown in the Figure 2 of the original manuscript derived from CD4^+^ T cells, whereas concentration shown in Figure 6 were measured in CD8^+^ T cells. We have now analysed ceramide contents in CD4^+^ T cells as well as CD8^+^ T cells isolated from both Asm^flox/flox^/CD4cre and Ac^flox/flox^/CD4cre mice included the new data in Figures 3, 6 and Supplemental Figures S2 and S3 of the revised manuscript.

6. Changes in ceramide levels may influence levels of other lipids, e.g., sphingolipids,glycosphingolipids, cholesterol, some specific glycerophospholipids (PS, PI). Do authors have any data about other lipids in Asm and Ac KO cells?

We agree with the reviewer that analysis of other lipids in AsmKO and AcKO T cells would be interesting. However, the focus of this study was on the effects of Asm deficiency and Ac deficiency on ceramide levels and the resulting phenotypic changes of T cells. Therefore, we do not have analysed other lipids yet. Nevertheless, in response to the suggestion of reviewer #1 we analysed S1P levels of Asm- and Ac-deficient T cells, but concentrations were under detection limit.

Minor comments:1. On 2-3 places, overstatement can be found: e.g., line 272 impaired -> reduced; line 453 diminished -> reduced/lowered

We have now carefully rephrased the respective statements.

2. Typo? Line 303: CD4^+^ T cells -> Tregs (not sure)

We thank the reviewer for this comment and specified our statement.

3. Figure 5B, granzyme B panel: The word 'Tumors'below the graph is confusing.

We apologize for this confusion. We have now corrected the Figure in the revised manuscript.